# Alternative Evolutionary Pathways in *Paspalum* Involving Allotetraploidy, Sexuality, and Varied Mating Systems

**DOI:** 10.3390/genes14061137

**Published:** 2023-05-24

**Authors:** Mara Schedler, Anna Verena Reutemann, Diego Hernán Hojsgaard, Alex Leonel Zilli, Elsa Andrea Brugnoli, Florencia Galdeano, Carlos Alberto Acuña, Ana Isabel Honfi, Eric Javier Martínez

**Affiliations:** 1Instituto de Botánica del Nordeste (IBONE-UNNE-CONICET), Facultad de Ciencias Agrarias, Universidad Nacional del Nordeste (FCA-UNNE), Corrientes 3400, Corrientes, Argentina; schedlermara@gmail.com (M.S.); vreutemann@gmail.com (A.V.R.); azilli@agr.unne.edu.ar (A.L.Z.); abrugnoli@agr.unne.edu.ar (E.A.B.); galdeanoflorencia@gmail.com (F.G.); cacuna@agr.unne.edu.ar (C.A.A.); 2Taxonomy & Evolutionary Biology, Leibniz Institute of Plant Genetics and Crop Plant Research (IPK), 06466 Gatersleben, Germany; hojsgaard@ipk-gatersleben.de; 3Programa de Estudios Florísticos y Genética Vegetal, Instituto de Biología Subtropical (PEFyGV, IBS-UNaM-CONICET), Posadas 3300, Misiones, Argentina; ahonfi@gmail.com

**Keywords:** apospory, cytogeography, fertility, mating systems, polyploidy, reproductive behavior, self-incompatibility

## Abstract

The genetic systems of *Paspalum* species have not been extensively studied. We analyzed the ploidy, reproductive mode, mating system, and fertility of four *Paspalum* species—*Paspalum durifolium*, *Paspalum ionanthum*, *Paspalum regnellii*, and *Paspalum urvillei*. An analysis of 378 individuals from 20 populations of northeastern Argentina was conducted. All populations of the four *Paspalum* species were pure tetraploid and had a sexual and stable reproductive mode. However, some populations of *P. durifolium* and *P. ionanthum* showed low levels of apospory. Populations of *P. durifolium* and *P. ionanthum* had low seed sets under self-pollination but were fertile under open pollination, showing that self-incompatibility likely caused self-sterility. In contrast, populations of *P. regnellii* or *P. urvillei* showed no evidence of apospory, and seed sets in both self- and open pollination conditions were high, suggesting that they are self-compatible due to the absence of pollen–pistil molecular incompatibility mechanisms. The evolutionary origin of the four *Paspalum* species could explain these differences. This study supplies valuable insights into the genetic systems of *Paspalum* species, which could have implications for their conservation and management.

## 1. Introduction

Polyploidy is a widespread phenomenon among species that entails a considerable influence on the evolution of plants, animals, and fungi [1]. Between 30 and 80% of angiosperms are polyploids [2]. Polyploidy acts as a primary mechanism for plant adaptation and speciation [3,4,5,6] by promoting variability through changes in chromosome numbers [7], increased allelic diversity in allopolyploids [8], and genomic restructuring [9]. In most cases, polyploidy poses an almost instantaneous and effective barrier to gene flow among cytotypes [10], allowing the coexistence of different intraspecific cytotypes [11]. Additionally, polyploid cytotypes generally differ from their diploid ancestors in morphological, ecological, physiological, reproductive, and cytological characteristics [7,12,13,14], contributing to the occupation of new ecological niches and the reinforcement of reproductive isolation mechanisms.

*Paspalum* L. is a large genus of Panicoid grasses, including around 310–350 species occurring mainly in the Americas [15,16]. The genus is distributed throughout tropical, subtropical, and temperate regions [17] with distinct species occupying flooded, dry, saline, or sandy soils of savannas, coastal dunes, tropical or temperate forests, and prairies [18]. The center of origin and diversification of the genus is tropical South America, with biodiversity hotspots in the Brazilian Cerrado and the Campos of Argentina, Uruguay, and southern Brazil [15,17]. The most common basic chromosome number in *Paspalum* is *x* = 10. However, a few exceptions to this number have been reported, such as species with *x* = 6 [16,19], *x* = 9 [20], and *x* = 16 [16]. 

*Paspalum* species show a broad range of genetic, morphological, and ecological diversity. Ortiz et al. [21] classified 72 *Paspalum* species into eight diverse groups based on their genetic system, which includes combinations of basic chromosome number, ploidy, reproductive modes, and mating systems. Approximately 75% of these species belong to groups that have polyploid cytotypes [21]. The most significant group includes species with sexual diploid cytotypes that are allogamous by self-incompatibility and apomictic tetraploid cytotypes from autopolyploid origin. Two other important groups lack diploid cytotypes and have an allopolyploid origin. One group is characterized by the exclusive occurrence of sexual polyploids (either 4*x* or 6*x*), and the other group has sexual tetraploids together with higher polyploid apomictic cytotypes [21].

Apomixis, the formation of asexual seeds, has been considered a “blind alley” or an evolutionary “dead end” [3,22], as apomictic plants lack mechanisms that exploit genetic variation during offspring formation, increasing the developmental instability of the sexual pathway and reducing the survival ability of sexual progenies [14,23,24,25]. Nevertheless, recent population studies have proved considerable amounts of genetic diversity in natural asexual populations [26]. Similarly, genetic variability studies in several natural populations of apomictic species of *Paspalum* showed elevated levels of genotypic variation, which have been attributed to backcrossing with sexual parental relatives, facultative sexuality, mutations, and the recurrent origin of new polyploids from divergent sexual ancestors [26,27,28,29,30,31].

Sexuality and hybridization have been significant factors in the origin and evolution of several *Paspalum* species. About 15% of *Paspalum* species are sexual tetraploids, all of which have an allopolyploid origin [21,32]. However, there is still a need for the comprehensive analyses of genetic systems, cytotype composition at the population level, and reproductive variability. Four species, belonging to distinct infrageneric groups being the main morphological variants within the genus [17], are particularly relevant in this regard.

*P. durifolium* Mez and *P. ionanthum* Chase are two species with multiple polyploid cytotypes that lack diploids [21]. However, the available records of ploidy and mode of reproduction are limited to single individuals [33,34,35,36,37,38]. Additionally, earlier studies of reproductive analysis relied on cytoembryology, and no studies of functional reproductive pathways using flow cytometric seed screening have been conducted.

*P. durifolium* is a tall, robust bunchgrass with coarse leaves that grows in lowlands near rivers and streams, and sometimes in grasslands in northern Uruguay, southern Brazil, northern Argentina, and Paraguay [17,33]. It has sexual tetraploid and apomictic hexaploid cytotypes with an allopolyploid origin [33,38,39]. A rare pentaploid plant reproducing by facultative apomixis in a tetraploid population [35] suggests cytotype variability within populations exists and points to the occurrence of dynamic population-level processes.

*P. ionanthum* is a warm-season perennial bunchgrass that grows in sandy and flooded soils in savannas of central Paraguay, Brazil, Argentina, and Uruguay [17]. This species has self-sterile sexual allotetraploid and apomictic octoploid cytotypes [34,37,38,40,41].

*P. regnellii* Mez and *P. urvillei* Steud. are species in which only sexual allotetraploid individuals have been found [24,42]. *P. regnellii* is found in southern and southeastern Brazil, eastern Paraguay, and northeastern Argentina, while *P. urvillei* is a warm-season perennial bunchgrass native to southern Brazil and northern Argentina, but it also grows from southern USA to Argentina [17]. Both species are commonly found in modified soils on the edges of forests and roads [17].

The knowledge of the geographic occurrences of cytotypes in different *Paspalum* species is yet fragmentary. However, in almost all cases, this information only relates to the ploidy level of a single isolated individual or germplasm collections (e.g., [37,40,43,44,45,46]). Although there have been a few extensive studies on cytotype diversity and distribution, they mainly focus on apomictic species (i.e., for *Paspalum intermedium* Munro ex Morong & Britton [13], for *Paspalum alcalinum* Mez, *Paspalum denticulatum* Trin., *Paspalum lividum* Trin., *Paspalum nicorae* Parodi and *Paspalum rufum* Nees [29], and for *Paspalum simplex* Morong [30,47]). As a result, there is a lack of information about reproductive variability and cytotype associations at local and regional scales, which is crucial for understanding the origin, dynamics, and adaptation of cytotypes in natural populations.

To address this gap, the present study aims (i) to sample a representative number of individuals from different natural populations of *P. durifolium*, *P. ionanthum*, *P. regnellii* and *P. urvillei*, (ii) to analyze their cytotype composition, (iii) to assess the reproductive mode at two developmental stages (i.e., ovules and seeds) under common environmental conditions, and (iv) to determine the mating system and fertility for each species. This information will shed light on the role of different genetic systems in the establishment of new cytotypes and their stability in the evolution of *Paspalum* agamic complexes.

## 2. Materials and Methods

### 2.1. Plant Collections

Plant materials were collected from the northeastern region of Argentina, which lies between the parallels of 26°24.117 S–31°01.223 S and parallels 54°14.845 W–59°25.005 W (Table 1). Populations of *P. durifolium* and *P. ionanthum* were collected near the Iberá wetland macrosystem (Corrientes, Argentina). Populations of *P. regnellii* were collected in Misiones (Argentina), while the populations of *P. urvillei* were collected in Corrientes, Chaco, Entre Ríos, Misiones, and Santa Fe (Argentina).

Sampling was conducted during two different periods—November–December for *P. durifolium* and December–February for *P. ionanthum*, *P. regnellii*, and *P. urvillei*. The sampling sites were chosen to ensure maximum representation of the distribution range of the species in the study region, considering macro-scale (among populations) and micro-scale (within populations) trends. The populations were at least 50 km apart from each other and exceeded a linear size of 200 m. Uniform sampling was completed to ensure an even representation of individuals within each population (i.e., the distance between two consecutive individuals was kept constant around 10 m). Five populations and ca. 20 plants per population were collected for each species.

Small cuttings with short rhizomes in the base were collected from each plant and cultivated in pots in a greenhouse. Once rooted, the plants were transferred to an experimental field near the city of Corrientes, Argentina. Herbarium vouchers were collected for each sampled population and deposited at the CTES and MNES Herbaria. The distribution of the ploidy levels of the analyzed populations were plotted on maps using DIVA-GIS [48]. Field GPS was used to decide the collection sites.

### 2.2. Ploidy Level Estimation

To determine the ploidy level of each sample, flow cytometry (FC) was used to estimate the relative nuclear DNA content in comparison with a plant with a known ploidy level of 2*n* = 4*x* = 40. The plants used for comparison by FC were *P. durifolium* Q3947 [39], *P. ionanthum* Q3726 [40], *P. regnellii* V10121 and *P. urvillei* Q4111 [49]. The ploidy level of each plant was determined using fresh leaf tissue samples following the instructions described in Brugnoli et al. [47]. The fluorescence intensity of 4′,6-diamidino-2-phenylindole (DAPI)-stained nuclei was analyzed with a Partec PA II flow cytometer (Partec GmbH, Münster, Germany) using the CyStain UV Precise P kit from Partec. The measurements of relative fluorescence intensity of stained nuclei were performed on a linear scale using the software PA I FloMax version 2.8.1 (Quantum Analysis GmbH, Münster, Germany), with at least 3000 nuclei per sample and CV ≤ 8%. The ploidy level of each sample was estimated from the histograms as a ratio between the mean value of the DNA peaks of the sample and the specific standard used.

For chromosome counting, a random sample of root tips was collected from potted plants, placed in a saturated solution of α-bromonaphthalene for 2 h, hydrolyzed in 1.0 mol m^−3^ hydrochloric acid for 10 min at 60 °C without previous fixation, and stained with Feulgen’s reagent. The root tips were macerated in a drop of 2% aceto-orcein stain, squashed under a coverslip, and observed using light transmission microscopy.

### 2.3. Reproductive Mode

To assess the reproductive pathways at two developmental stages of the life cycle in populations of four *Paspalum* species, two analytical methods were employed. The first method involved cyto-embryological analysis of mature ovules, while the second employed flow cytometric seed screening (FCSS).

Cytoembryological analysis by mature embryo sac observation was performed on a random sample of five plants from each population. Spikelets at anthesis were fixed in FAA (70% ethanol, 37% formaldehyde, and glacial acetic acid in the ratio 18:1:1). Pistils were dissected out and clarified following the method described by Young et al. [50] with the modifications carried out by Zilli et al. [51]. Between 30 and 40 pistils were observed per plant, using a Leica DM2500 (Leica, Wetzlar, Germany) with a Nomarski differential interference contrast device. Embryo sacs were photographed with a Leica EC3 camera. Based on the structure and cell composition of the embryo sacs, four types of ovules were defined: (i) ovules carrying one meiotic embryo sac (MESs) composed of an egg cell, two synergid cells, a central cell with two (rarely one or three) nuclei, and several antipodals; (ii) ovules carrying one or more aposporous embryo sacs (AESs) with an egg cell, one or two synergid cells, a central cell with two (rarely one or three) nuclei, and no antipodals; (iii) ovules with mixed embryo sacs having one MES plus one (or more) AES; and (iv) ovules without embryo sacs (AbESs) whereby embryo sacs were not observed or had clear signs of abortion. Plants bearing only ovules with one meiotic embryo sac (MES) were classified as sexual, while those with both ovules with MES and one or more aposporous embryo sacs (AESs) were classified as facultative sexual. 

The FCSS method was used to reconstruct the reproductive pathways of fully developed caryopses (seeds). Seeds from open-pollinated flowers of five plants per population were analyzed using FCSS according to protocol described by Matzk et al. [52] and following instructions described by Siena et al. [53]. Seeds were harvested from the same plants employed for cyto-embryology. Thirty seeds per individual were analyzed in bulks of five seeds each. The fluorescence intensity of 4′,6-diamidino-2-phenylindole (DAPI)-stained nuclei was determined using a Partec PA II Flow Cytometer (Partec GmbH, Münster, Germany) with the detector running at 355 nm. Ploidy levels of the endosperm and embryo tissues were estimated by comparing the different peak configurations. A flow cytometry histogram with two main peaks equivalent to values of 2C and 3C would indicate that the seed was formed sexually (embryo = *n* + *n*; endosperm = *n* + *n* + *n*, i.e., meiosis + syngamy). Alternatively, a histogram showing two main peaks corresponding to 2C and 5C values would reveal an apomictic origin of the seed (embryo = 2*n* + 0; endosperm = 2*n* + 2*n* + *n*, i.e., apospory + parthenogenesis + pseudogamy). Approximately 3000 nuclei were measured per sample, and data analysis was performed using the software PA I FloMax version 2.8.1 (Quantum Analysis GmbH, Münster, Germany). The mean values of relative DNA content (C-values) for the embryo and endosperm tissues of single seeds were set up to infer the sexual or apomictic origin of each seed.

### 2.4. Reproductive Pathway Efficiency

The efficiency of the reproductive pathways (sexual and apomictic) was evaluated by calculating the ratio between the expected and observed frequencies of ovules undergoing the meiotic or apomictic pathway at each stage, as described by Hojsgaard et al. [23] and Reutemann et al. [54]. At the ovule stage, the proportion of embryo sacs was estimated by *nm*/*nt* for the meiotic pathway and *na*/*nt* for the apomictic pathway, where *nm* is the total number of ovules with a meiotic embryo sac (MES), *na* is the total number of ovules with apomictic embryo sacs (AES), and *nt* is the total number of embryo sacs. The values of *nm* and *na* include the number of observed ovules with mixed embryo sacs (MES + AES), since both reproductive pathways were expressed. At the seed stage, the observed proportions of each reproductive pathway were calculated based on the number of sexual and apomictic seeds. The expected proportions of each pathway at the seed stage were determined by using the observed proportions of the two reproductive pathways at the ovule stage.

To evaluate the differences between the observed proportions of both pathways in each stage, a paired *t*-test was conducted on the mean difference. Additionally, a standard Pearson’s Chi-squared test was performed to find significant differences between the expected and observed proportions of each reproductive pathway. 

### 2.5. Mating System and Seed Fertility

The mating system and seed fertility were examined by analyzing the seed set during two consecutive flowering periods (2015–2016 and 2016–2017) under conditions of forced self-pollination and open pollination. Random samples of five plants per population were analyzed to determine the mating system. The degree of autogamy and the breaking of self-incompatibility were estimated by analyzing the proportions of seed set in self-pollination. To ensure self-pollination, three inflorescences per individual plant were bagged before anthesis. The degree of allogamy was estimated by comparing the seed set under open pollination and self-pollination. Additionally, three inflorescences per individual were bagged after anthesis, and spikelets with a caryopsis were separated from empty spikelets using a seed blower. After approximately one month, inflorescences were manually threshed, and the seed set was calculated using the formula: seed set = (no. of seeds/total no. of spikelets) × 100 for each pollination condition and plant.

### 2.6. Statistical Analysis

The seed set data were analyzed using the Info-Gen software version 2016 [55] in a completely randomized design where each plant stood for a replication in the population. Statistical comparisons for seed set were performed in the following sequence: (i) comparisons within populations between the two flowering periods (1st vs. 2nd) for each pollination method (self- and open pollination), (ii) comparisons among populations for each pollination method (self- and open pollination) and flowering period (1st and 2nd), (iii) comparisons within populations between the pollination methods (self- vs. open pollination), based on the overall mean values of both flowering periods, and (iv) comparisons among populations for each pollination method (self- and open pollination), based on the overall mean values of both flowering periods. Mean values, coefficients of variation, and ANOVA were computed. Tukey’s test was employed ad hoc when ANOVA showed significant differences in the mean comparisons. Unless otherwise specified, all differences were considered significant at *p* < 0.05.

## 3. Results

### 3.1. Ploidy Estimations

All individuals from populations of *P. durifolium* were tetraploids (2*n* = 4*x* = 40). Ninety-two individuals analyzed showed 2C DNA content similarly to the tetraploid standard of *P. durifolium* (Figure 1A). No hexaploid or pentaploid individuals were detected in the populations of this species. Similarly, all 96 plants from five populations of *P. ionanthum* were tetraploids (2*n* = 4*x* = 40) (Figure 1B), and no expected octoploid plants or plants with other ploidy levels were detected. The populations of *P. regnellii* and *P. urvillei* also showed tetraploid individuals with all 96 plants from *P. regnellii* and 94 plants of *P. urvillei* being tetraploid (2*n* = 4*x* = 40) (Figure 1C,D). These results were further confirmed by chromosome counting in root tip meristems for a random sample of four plants from each population (Figure 2).

### 3.2. Embryo Sac Analysis

Embryo sac analysis was conducted on one hundred plants from all populations of the four species under study. The findings, presented in Table 2, summarized the types of embryo sacs observed in each species and population. Among the populations of *P. durifolium*, a high proportion of ovules with well-developed embryo sacs were observed (Table 2 and Appendix A). Twenty-five evaluated plants from five *P. durifolium* populations showed ovules mainly with MES, ranging between 79.5 and 94% (Table 2). These meiotic sacs are of *Polygonum* type, consisting of an egg cell, two synergids, a central cell with two polar nuclei, and three antipodal cells, which later divide into several antipodal daughters (Figure 3A). The highest percentage of ovules with meiotic embryo sacs (MESs) was observed in population PD2, and no AES or MES + AES were observed (Table 2 and Appendix A). However, in the remaining four populations, some plants showed ovules with MES next to an AES carrying an egg cell, one or two synergids, and a large binucleate cell, lacking antipodal cells (Table 2, Figure 3E). The population PD4 had the highest mean percentage (6.5%) of ovules with MES + AES (Table 2). Populations PD3 and PD4 had at least one plant showing near 20% of MES + AES ovules type (Appendix A). Population PD5 had the highest proportion of individuals with ovules carrying MES + AES (3 out of 5; Appendix A) and a mean percentage of ovules with mixed embryo sacs of 3.2% (Table 2). No ovule carrying only AESs was observed. AbESs were also present in all populations, ranging between 6 and 17.3% (Table 2). Five populations showed significant differences with the meiotic pathway dominating at the ovule stage (*p* < 0.001, Table 2).

In *P. ionanthum*, the analysis of embryo sacs among 25 individuals from all populations revealed a high percentage of ovules with viable embryo sacs (Table 2 and Appendix A) with most individuals carrying one MES in a high proportion of ovules (Figure 3B). Populations PI1 and PI2 showed 2.5 and 1.3% of ovules with MES + AES, respectively (Table 2). Such ovules were observed in two individuals from PI1 and one from PI2. Ovules with a single AES were not observed (Appendix A). Significant differences between both reproductive pathways were observed in the five populations (*p* < 0.001, Table 2), with meiotic pathway dominating at the ovule stage.

Populations of *P. regnellii* showed a high proportion of ovules with well-developed embryo sacs (Table 2 and Appendix A), with ovules with MES (Figure 3C) ranging between 70.0 and 95.5% in all populations (Table 2). The population PR3 showed the highest percentage of ovules with MES, while PR1 showed the lowest percentage due to a high proportion of ovules with AbESs (Table 2). No aposporous embryo sacs were observed in any of the twenty-five analyzed plants (Table 2 and Appendix A). Five populations showed significant differences, with the meiotic pathway as the only one expressed at the ovule stage (*p* < 0.001, Table 2).

Populations of *P. urvillei* showed the highest percentage of ovules with MES (Figure 3D), ranging between 95.8 and 100% among populations (Table 2), with all evaluated plants from population PU5 and four out of five individuals from population PU3 showing ovules with 100% MESs (Appendix A). No AES were detected in the ovules of the 25 analyzed plants of *P. urvillei*, and the average percentage of ovules without embryo sacs was 4.3% in PU1 (Table 2). The meiotic pathway was the only one expressed at the ovule stage (*p* < 0.001, Table 2). 

### 3.3. Flow Cytometric Seed Screen

Seeds from each population of the four species were analyzed by Flow Cytometric Seed Screen (FCSS). Specifically, 150 seeds from five plants of each population were assessed (Table 2), and the FCSS analysis revealed a single type of histogram peak for all individuals in each population. This peak corresponded to the embryo: endosperm ratios of relative DNA content 2C:3C (Figure 4). Furthermore, histograms from cell nuclei of bulked seed samples of *P. durifolium* were examined, and four distinct peaks were observed (Figure 4). The two highest peaks corresponded to stages G1 (4C) and G2 (8C) of tetraploid embryos, while the peaks at 150 and 300 of relative DNA content corresponded to stages G1 (6C) and G2 (12C) of the endosperms. Additionally, two smaller peaks were observed at 400 and 600 of relative DNA content, which could indicate stages G2 of an embryo (16C) and endosperm (24C) from a hexaploid seed, respectively. The peaks between 100 and 300 of relative DNA content suggested the presence of sexual reproduction, including a reductional division (meiosis) and double fertilization. However, the peaks at 400 and 600 showed double fertilization from AES (3C:5C) rather than typical sexual reproduction. Notably, the study did not observe any instances of seed originating from apospory + pseudogamy (Table 2).

### 3.4. Reproductive Variability in Ovules and Seeds

The proportions of sexuality and apomixis during the reproductive stages displayed distinct patterns in *P. durifolium* populations. Populations PD1, PD3, PD4, and PD5 showed a low proportion of the apomictic pathway during the ovule stage, but all populations of *P. durifolium* produced 100% sexual seeds during the seed stage (Table 3). Furthermore, there were no significant differences between the observed and expected proportions of sexually originated seeds (*p* > 0.05, Table 3). The efficiency of the sexual pathway from the ovule to the seed stage ranged from 1.01 to 1.07, showing an increase in the sexual pathway (Table 3). 

Similarly, in populations PI1 and PI2 of *P. ionanthum*, which showed a low proportion of the apomictic pathway during the ovule stage, only sexual seeds were produced (Table 3). In five populations of *P. ionanthum*, there was a high efficiency of the sexual pathway from the ovule to the seed stage, with values ranging from 1.0 to 1.03 (Table 3).

In *P. regnellii* and *P. urvillei*, only the sexual pathway was expressed during both reproductive stages. Both species showed a high degree of efficiency of the sexual reproductive pathway (Table 3). 

### 3.5. Mating System and Seed Fertility 

The intra-population seed set of *P. durifolium* under forced self-pollination conditions ranged from 0.4 to 0.9% based on the overall mean values, while under open pollination conditions, it ranged from 28.8 to 51.5% (Table 4). Population PD3 exhibited the lowest and highest seed sets under self- and open pollination, respectively. The seed production under self-pollination was highly variable in both flowering periods, with a high and wide population coefficient of variation (CV). A lower CV was observed under open pollination (Appendix A). There were no significant intra-population differences (*p* > 0.05) observed in either self- or open pollination when comparing the two flowering periods. Inter-population significant differences were only observed in open pollination (*p* < 0.006) when considering the overall mean values. The fertility was significantly higher under open pollination than in self-pollination (*p* < 0.001) (Table 4).

In *P. ionanthum*, the intra-population seed set under forced self-pollination conditions ranged from 0.6 to 2.8% based on the overall mean values, while under open pollination, it ranged from 29.4 to 54.3% (Table 4). Population PI3 exhibited the highest values in both self- and open pollination, reaching up to 4.26% in self-pollination during the first flowering period (Table 4 and Appendix A). The seed set in self-pollination was highly variable in both flowering periods. A lower CV was observed in open pollination (Appendix A). However, regarding intra-population pollination, there were no significant differences between the two flowering periods for both pollination conditions, except for PI2 at open pollination (*p* < 0.002). As for inter-population pollination, significant differences were observed in open pollination during the second flowering period (*p* < 0.004) and when considering the overall mean values (*p* < 0.017). The fertility in open pollination was significantly higher than in self-pollination (*p* < 0.001) (Table 4).

In *P. regnellii*, the intra-population seed set under forced self-pollination conditions ranged from 6.4 to 18.8% based on the overall mean values, while in open-pollination conditions, it ranged from 26.4 to 49.8% (Table 4). Population PR5 showed the highest values in both self- and open pollination. The seed set in self-pollination for the two flowering periods showed a high and wide CV, while in open pollination, it was lower (Appendix A). Populations did not show significant intra-population differences between the two flowering periods for both pollination conditions except for PR2 in self-pollination (*p* < 0.018). Inter-population significant differences were observed in open pollination during the first flowering period (*p* < 0.006). In turn, there were also significant differences among populations, both in self-pollination (*p* < 0.039) and open pollination (*p* < 0.026), when considering the overall mean values. The fertility in open pollination was significantly higher than in self-pollination (*p* < 0.001 to 0.031) (Table 4). 

The seed set within populations of *P. urvillei* under forced self-pollination showed a range of 35.6 to 43.1% based on the overall mean values. Conversely, in open pollination, the range was 70 to 80.6%, as shown in Table 4. Among the populations, PU5 and PU2 displayed the highest values in self- and open pollination, respectively. The coefficient of variation (CV) for the seed set in self-pollination was higher than that in open pollination (Appendix A). Notably, only population PU3 exhibited significant differences in open pollination between the two flowering periods (*p* = 0.007). When considering the overall mean values, there were no significant inter-population differences in either self- or open pollination during each flowering period. Fertility in open pollination was significantly higher than in self-pollination (*p* < 0.001 to 0.018) (Table 4).

## 4. Discussion

Polyploidy and apomixis are common in the *Paspalum* genus, with most species being tetraploid and having diploid sexual counterparts that are co-specific [21,32]. However, there is a subgroup of species within the genus that show a different genetic system, being both polyploid and sexual, and some of these also have apomictic co-specific counterparts of higher ploidy. Knowledge about the genetic system of these polyploid species has been limited until now, with most analyses being based on one or few individuals from separate collections. Population-level studies have been carried out only on apomictic species within the genus. However, a recent population analysis of four sexual polyploid species in *Paspalum* has allowed for a better understanding of their natural distribution and the variation in their genetic systems.

### 4.1. Single Ploidies Featured in All Species

The ploidy level of the five *P. durifolium* natural populations was tetraploid (2*n* = 4*x* = 40), and they were uniform with respect to chromosome number and cytotype composition. Although hexaploid and pentaploid individuals have been reported for the species by other authors [33,35,36,38,39], they were not detected in the present study. In all these cases, the records were of individual 5*x* or 6*x* plants, except for Quarin [39], who evaluated 12 plants belonging to 8 different accessions and found a frequency of 0.86 (10/12) 4*x* and 0.14 (2/14) 6*x*. All populations were found in association with the Iberá wetland macrosystem, which is found in Corrientes, Argentina. Quarin [39] mentions that the tetraploid cytotype of *P. durifolium* is found in the central part of the Province of Corrientes, which is associated with the Iberá basin, while the hexaploids are found toward the eastern end of Corrientes in the Uruguay River basin. Two tetraploid populations from this study (PD3 and PD4) were collected in Santo Tomé (Corrientes) near the Uruguay River basin where hexaploids were previously reported by Quarin [39]. The hexaploid cytotype was also registered in Rio Grande do Sul, Brazil [36,38] and in Rivera in the north of Uruguay [33]. We also did not find the pentaploid cytotype of *P. durifolium* in any of the wild populations evaluated, which could show the existence of hexaploids within the area. Natural pentaploid individuals have their origin in the hybridization between tetraploid and hexaploid cytotypes, and their persistence depends on the degree of sterility and the mode of reproduction, since both apomictic and sexual reproductive pathways are usually functional in this cytotype [35]. Our results show a wide distribution of the tetraploid cytotype of *P. durifolium* in its natural range at the northeast of Argentina, which may be associated with a higher chromosomic and reproductive stability. 

All five populations of *P. ionanthum* turned out to be pure tetraploids (2*n* = 4*x* = 40) in agreement with all previous records of individual chromosome counts for the species [34,37,38,40,41,42,56]. Populations were found in low and humid lands, on sandy and flooding soils, which are typical environments for this species [17]. No octoploid cytotype was detected as contrary to earlier reports by Burson and Bennett [34] and Pozzobon et al. [38,41]. Burson and Bennett [34] analyzed two plants belonging to two accessions, where one individual was found to be tetraploid and the other was found to be octoploid. Pozzobon et al. [38] and [41] observed a frequency of 0.11 (1/9) and 0.2 (2/5) octoploid individuals, respectively. The octoploid cytotype of *P. ionanthum* may have originated through a polyploidization mechanism described for other polyploid species of *Paspalum* [32]. Our results show that the incidence of such a mechanism in *P. ionanthum* and the generation of elevated level polyploids is not frequent, being the tetraploid cytotype the only one—currently—occurring in Argentina.

Ploidy analyses conducted on five populations of *P. regnellii* have confirmed that only the tetraploid cytotype exists in the species’ natural geographic range in Argentina. Previous records of individual plants from this species have also confirmed their tetraploid nature [24,38,43,45,57]. All populations were found in similar environments on modified red clay soils at the roadsides and edges of native or cultivated forests. The species’ natural distribution extends throughout the center and south of Brazil, the east and northeast of Paraguay, and the northeast of Argentina [17,24]. 

Populations of *P. urvillei* collected in northeast Argentina were exclusively found to be tetraploids, confirming earlier records that show the existence of only one cytotype [42,43,45,58,59,60,61]. A hexaploid reported by Nielsen [62] was represented by a plant collected in the city of Magnolia, Arkansas, United States. The age of this record and the lack of confirmation of its taxonomic identity cast doubts on the accuracy of such a register. The species is native to South America, inhabiting Bolivia, Brazil, Chile, Paraguay, Argentina, and Uruguay, where it grows in low fields, estuaries, and on the edges of roads and railways, and it is a common weed in crop fields [17]. 

### 4.2. Reproductive Pathways and Cytotype Stability

The analysis of reproductive modes in *Paspalum* species allows for the evaluation of sexual and apomictic pathways and an understanding of the efficiency of each pathway at different developmental stages [23,54]. In our study, we measured the mode of reproduction in two stages of the plant’s life cycle, the ovule and seed, to expose variations in the expression levels of sexuality and/or apomixis at the individual and population levels. This information can help us understand the cytotype dynamics in nature and predict possible ploidy compositions in different geographical areas. 

Populations of *P. durifolium* showed differences in reproductive pathways at the ovule stage, particularly during the observation of the mature female gametophyte. The meiotic pathway was exclusive in population PD2, with 94% of mature meiotic embryo sacs (MESs) being fully developed. The apomictic pathway was not expressed during ovule development in this population, as no aposporous embryo sacs (AESs) were observed among the 150 ovules analyzed. Conversely, in the remaining four populations, both reproductive pathways (sexual and apomictic) were observed with distinct levels of expression. In all of them, the meiotic pathway was expressed in a higher proportion than the apomictic one. The apomictic pathway was expressed in only 2.9% of the total ovules analyzed and was always accompanied by the expression of the sexual pathway in the same ovule. All ovules with MES + AES showed a greater development of the meiotic sac than the aposporous sac. This entails a competitive advantage favoring seed development from the MES over the AESs in mixed ovules. Quarin [39] saw a similar reproductive behavior in some tetraploid accessions of *P. durifolium*. Out of 79 mature ovules analyzed, 63 had a normal MES, seven had one MES plus one AES, one ovule showed two AESs, and eight had undeveloped embryo sacs. The functionality of these AESs has never been tested in *P. durifolium*. There is no evidence that the pentaploid or hexaploid cytotypes originated from these AESs for polyploidization. Cytological results proved that the apomictic 6*x* cytotype of *P. durifolium* was generated by allopolyploidy, and it has three different genomes [33]. Moreover, the pentaploid cytotype found in a tetraploid population probably had its origin via the fertilization of a reduced female gamete from a tetraploid plant (*n* = 20) with a reduced male gamete from a hexaploid progenitor (*n* = 30) [35].

A comparable reproductive behavior was observed in the mature ovule stage of *P. ionanthum* populations. Sexuality was the only reproductive pathway expressed in three populations (PI3, PI4, and PI5), with a potential for sexuality close to 100% due to the extremely small number of ovules without embryo sac development. Two populations (PI1 and PI2) showed both reproductive pathways (sexual and apomictic), with a low potential for apomixis since some ovules were discovered carrying both MESs and AESs. All ovules with MES + AES showed a well-developed MES and a smaller undeveloped AES. The presence of MES + AES in tetraploid *P. ionanthum* was also observed by Burson and Bennett [34] and Martínez et al. [40]. Nevertheless, seeds produced by parthenogenesis (apomixis) or the fertilization of unreduced female gametes from AESs have never been conclusively proved in *P. ionanthum*. The octoploid individual of *P. ionanthum* (syn. *P. guaraniticum* Parodi) was taxonomically accurately identified and, according to meiosis analysis, originated from autopolyploidy [34]. The chromosome duplication mechanism that led to this octoploid is still unknown; however, a possible origin through the fertilization of two unreduced gametes from tetraploid plants (2*n* = 4*x* = 40), where the female gametes could be from an AES, is well-supported. 

Populations of *P. regnellii* and *P. urvillei* displayed similar behavior in both reproductive stages, showing exclusively the sexual pathway. At the ovule stage, most populations of *P. regnellii* demonstrated a high potential for sexuality, and no evidence for vestigial apospory was detected. Population PR1 exhibited a lower potential for sexuality (70%) due to a high proportion of ovules with AbESs. Previously, the megasporogenesis and megagametogenesis of *P. regnellii* were evaluated in one accession by Norrmann [24] who saw only MESs of the *Polygonum* type. All populations of *P. urvillei* had at the ovule stage a potential for sexuality over 95%, because most ovules showed a well-developed MES. Evidence for apospory was not detected in our observations of *P. durifolium* and *P. ionanthum*. In *Paspalum*, polyploids without apomixis are extremely rare. In *P. regnellii,* a full expression of sexuality is associated with endemism; however, in *P. urvillei*, it is linked with the potential colonization of new suitable and hospitable habitats in disturbed ecosystems. 

Gametophytic apomixis is a trait often linked to polyploidy, as individuals with the potential for apomixis in tetraploid populations can give rise to new hexaploid cytotypes. In *Paspalum*, this can occur through the fertilization of unreduced gametes from an aposporous embryo sac of a tetraploid individual by a reduced male gamete of another tetraploid individual, as described by Quarin [32]. This polyploidization mechanism was also observed in mixed populations (2*x* and 4*x*) of *P. rufum*, where the unreduced female gametes from the AES of a diploid plant are fertilized by reduced male gametes of tetraploid plants to create new tetraploids [53]. 

While sporadic AESs going with MESs have been detected in several sexual diploid *Paspalum* species [37,53,63,64,65,66], the functionality of these AESs has only been proved in *P. rufum* [53]. In diploid genotypes of *P. rufum*, variations in the functionality of AESs and the ability for apomixis have been observed in self-pollination and interspecific interploidy crosses [53,67]. However, seed screen analyses performed for *P. durifolium*, *P. ionanthum*, *P. regnellii* and *P. urvillei* did not detect any seeds of apomictic origin, showing that these species reproduced solely through the sexual pathway. This could be due to the competition for resources or space within the ovule, temporal development, and even to a seasonal effect, as seen in other sexual species [64,67,68,69].

The use of flow cytometry seed screen analyses revealed that none of the populations of *P. durifolium*, *P. ionanthum*, *P. regnellii*, and *P. urvillei* produced seeds of apomictic origin. The analysis of the DNA content ratios of seed embryo and endosperm tissues showed that all seeds were generated through sexual reproduction. This finding is consistent with the results obtained by Galdeano et al. [49], who also used FCSS to study these four species and found that all seeds were produced through the sexual pathway. Although the tetraploid populations of *P. durifolium* and *P. ionanthum* possess the potential for apomixis at the ovule stage, this pathway is unexpressed at the seed stage, which is likely due to competition for resources or space within the ovule. This behavior has also been observed in other sexual species and was attributed to the competition for resources or space within the ovule, temporal development, and even to a seasonal effect [67,68,69]. Siena et al. [53] observed functional AESs in the diploid cytotype of *P. rufum*, where a small proportion of seeds developed from AESs. The genetic program necessary for the development of AESs appears to be present and expressed in the tetraploid populations studied here, but factors such as the functionality of the parthenogenesis gene(s), developmental competition between embryo sacs, and the genotype background may hinder the development of apomixis in these plants. Therefore, the reproductive behavior of these plants can only be considered as obligate sexual, and the possibilities for the development of apomixis remain elusive and complex. 

### 4.3. Two Mating Systems with Partial Breakdown of Self-Incompatibility

Populations of *P. durifolium* showed an extremely low seed set under forced self-pollination conditions, with an overall average seed set value of 0.58%. On the other hand, seed set under open pollination was 40.8% and showed significant differences between populations. Quarin [39] observed in a single tetraploid plant of *P. durifolium* a seed set under self- and open pollination of 0.26 and 68.5%, respectively. The lower fertility via self-pollination was attributed to the existence of a pollen–pistil self-incompatibility system, mainly supported by a lack of penetration of the pollen tube into the style, and the observation of successful cross-pollination between different tetraploid genotypes [39]. 

Most sexual diploid *Paspalum* species show self-incompatibility, which is the most common mechanism of pollen–pistil interaction in self-sterile diploids such as *P. fasciculatum* Willd. ex Flüggé [70], *P. notatum* Flüggé [71], *P. procurrens* Quarin [72], and *P. indecorum* [63]. Self-incompatible diploids generally have con-specific self-compatible apomictic tetraploids [21,32,63]. The breakdown of the self-incompatibility system in these *Paspalum* species is likely due to polyploidization [32,73], as has also been described in other species (revised by [74]). This mating system, in which self-sterile sexual diploids and self-fertile apomictic polyploids coexist, is seen in several *Paspalum* species, including *P. durifolium*, but at the polyploid level. Sexual tetraploids of *P. durifolium* are self-incompatible that exhibit allogamy [39], while hexaploids are self-compatible pseudogamous apomictic [33]. 

Fertility in *P. ionanthum* under forced self-pollination conditions was low, with an overall population mean of 1.4%, which is similar to that observed by Quarin and Norrmann [37] for a single plant. Two populations (PI3 and PI5) showed higher values (4.26 and 2.85%) during the first flowering period. The seed set in the first measured period was carried out during a warmer and drier summer than in the second period. In this species, there is a clear partial breakdown of the self-incompatibility system, and the molecular basis for it could be due to environmental stressors such as elevated temperature during pollination [75,76,77,78] or internal factors, either genetic or cytological [76,79,80]. Wilkins and Thorogood [75] showed that self-incompatibility is broken in plants of perennial ryegrass (*Lolium perenne* L.) after exposing maturing anthers to 34 °C prior to anthesis, thus increasing the seed set. On the other hand, Hirosaki and Niikura [76] observed in *Brassica rapa* L. that the level of self-incompatibility decreases when maximum temperatures are >26 °C and plants are in the last quarter of the flowering period. The factors influencing shifts in the level of self-incompatibility varied across genotypes, depending upon the flowering period, plant senescence, and average temperature prior to pollination [76,81]. Our results show that all populations of *P. ionanthum* had a higher fertility under open pollination conditions with an overall average value of 42.4%. Quarin and Norrmann [37] obtained a seed set of 70% under open pollination conditions. Hence, we confirm that the tetraploid cytotype in *P. ionanthum* is mostly self-sterile due to a system of self-incompatibility acting during the pollen–pistil interaction, and therefore, it behaves as an allogamous species. Our observations also suggest that this self-incompatibility mechanism can be partially broken, depending on internal and external factors, but its influence on the seed set still is to be determined. 

Populations of *P. regnellii* produced seeds through both self- and open pollination, although the respective percentages varied both within and between populations. In self-pollination, the overall population mean was 11.8%, with significant differences observed among populations. Conversely, in open pollination, the overall population mean was 34.9% without inter-population differences. Norrmann [24] reported seed set percentages of 80 and 84% in self- and open pollination, respectively. The lower fertility observed in this study could be attributed to genetic and/or population-level variations, random effects caused using different sulfite paper envelopes, and/or variations in environmental conditions during data collection. *Paspalum regnellii* typically grows on the edge of forests or pine plantations under semi-shaded conditions on wet lateritic red soils with high iron content [17]. However, our evaluations were conducted on plants grown in the experimental field of the Facultad de Ciencias Agrarias (FCA-UNNE) in Corrientes, Argentina, which had conditions that differed significantly from the ideal natural settings. Likewise, Norrmann [24] conducted measurements on a few plants grown in pots with lateritic red soil, which were kept inside a greenhouse with controlled humidity and temperature conditions. External environment and the maintenance of generous size inflorescences enveloped for several days under natural climatic conditions possibly had a greater influence on the lower seed production observed both in self- and open pollination. Nonetheless, the self-fertility values observed in this study supply evidence of the selfing behavior of *P. regnellii*. 

Similarly, *P. urvillei* exhibited a similar behavior to *P. regnellii* regarding its fertility, producing seeds under both self- and open pollination but with lower values when forced to self-pollinate. In self-pollination, the overall population mean was 41.4%, while in open pollination, it was 74.4%. These values were lower than those observed by Caponio and Quarin [82] in a single *P. urvillei* plant analysis, where they reported seed sets of 75.0% and 84.4% under self- and open pollination, respectively. The lower fertility observed in this study could be due to the same causes mentioned for *P. regnellii* with a greater effect on self-pollination due to the longer enveloped period. The self-fertility values saw in this study also corroborate the selfing behavior of *P. urvillei*. 

The correlation between polyploidy and self-fertilization is widely recognized among many flowering plants [83,84], and in some species of *Paspalum*, a breakdown of self-incompatibility mechanisms has been seen [21]. Selfing is often found in pioneer plants, weeds, and island endemics [85,86]. *Paspalum regnellii* and *P. urvillei* are pioneer species that typically grow in anthropic environments on the edge of forests and roadsides, where selfing is a favorable condition for their local establishment and has likely been selected by evolution. Self-fertilization systems have probably evolved early among angiosperms, requiring the breakdown of self-incompatibility systems to produce seeds [74]. Self-incompatibility causes self-sterility by preventing pollen grains from adhering to the stigma or inhibiting pollen tubes from growing down the style [74,87,88], and it is typically controlled by the monogenic or multigenic control of S alleles [89,90,91]. In allotetraploid species such as *P. durifolium* and *P. ionanthum*, a multiple-locus system may not eliminate the self-incompatible alleles, but it can widen allele diversity and create dosage effects. Compared to a single-allele system that acts on the interaction between haploid gametes, polyploid gametes will have at least two alleles interacting with each other, helping the breakdown of incompatibility. Alternatively, in allotetraploid species such as *P. regnellii* and *P. urvillei*, the lack of self-incompatibility may be selected by ecological constraints and the evolutionary history of these two species, perhaps originating from parents with a multigenic system, making it difficult to maintain a gene-mediated self-incompatibility system.

## 5. Conclusions

Our study presents a comprehensive analysis of the genetic systems of four polyploid *Paspalum* species at the population level. Our results show a stable occurrence of a single ploidy in all species and populations, with sexual, highly fertile tetraploids showing alternative pollination syndromes. However, we saw reproductive variability among individuals and populations and the occurrence of alternative sexual and apomictic pathways in *P. durifolium* and *P. ionanthum*. Although the apomictic pathways were non-functional in our population-level screenings, the study highlights the potential role of unreduced female gametophytes in the formation of new polyploid cytotypes not previously recorded for these species.

Our study also showed that autogamous self-compatible and allogamous self-incompatible tetraploids were equally fertile under open pollination. Nevertheless, the partial breakdown of the self-incompatibility system in *P. regnellii* and *P. urvillei* suggests a transition from allogamy to autogamy that may be influenced by the species’ particular ecological niches.

These findings have significant implications for biodiversity conservation plans and the management of genetic resources as well as for genetic improvement programs. Further genetic diversity analysis will supply insights into the evolutionary role of mating systems and polyploidy in the maintenance, dispersal, and adaptation of these species, and it will support the development of effective conservation strategies.

## Figures and Tables

**Figure 1 genes-14-01137-f001:**
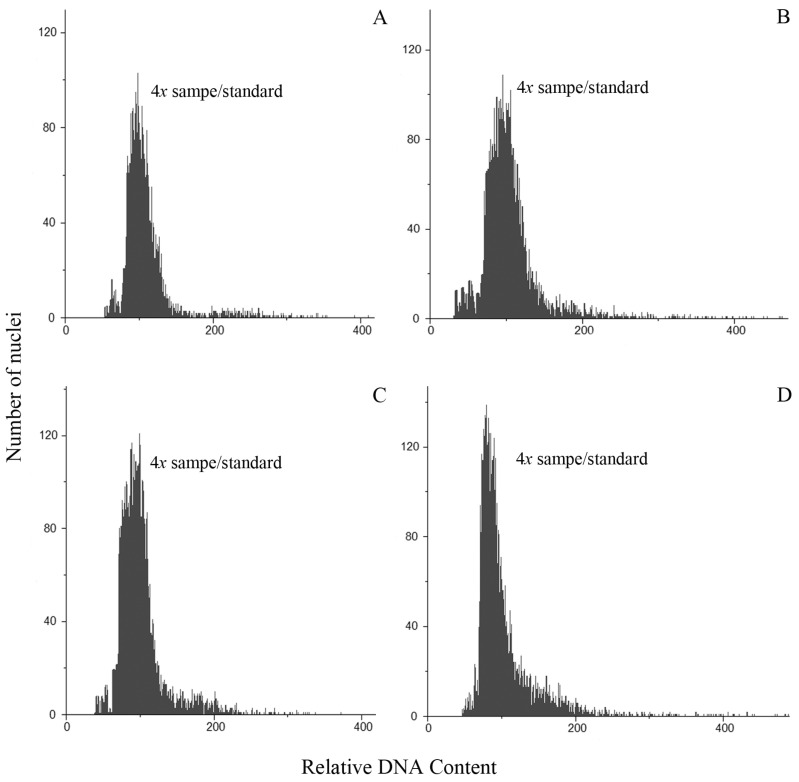
Histograms of cell nuclei from leaves of a sample of unknown ploidy and the standard sample for *P. durifolium* (**A**), *P. ionanthum* (**B**), *P. regnellii* (**C**), and *P. urvillei* (**D**). In all cases, the ploidy of the unknown sample is the same as the tetraploid (2*n* = 4*x* = 40) standard used for each species.

**Figure 2 genes-14-01137-f002:**
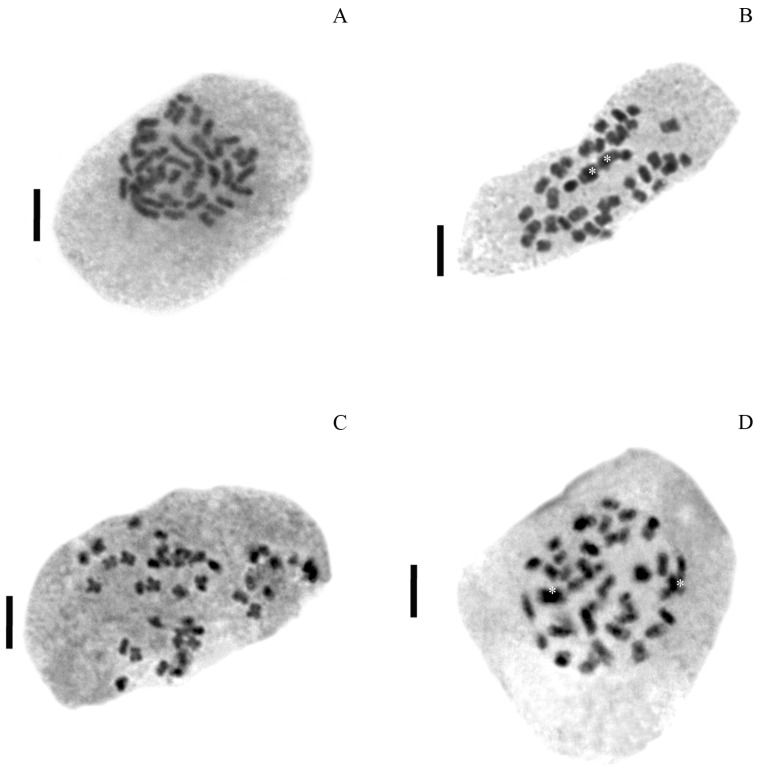
Chromosome counts in four Paspalum species. Mitotic metaphase showing 2*n* = 40 chromosomes in *P. durifoloium* PD5 (**A**), *P. ionanthum* PI4 (**B**), *P. urvillei* PU1 (**C**) and *P. regnellii* PR1 (**D**). Bar scale: 5 μm. Asterics in (**B**,**D**) show superposed chromosomes.

**Figure 3 genes-14-01137-f003:**
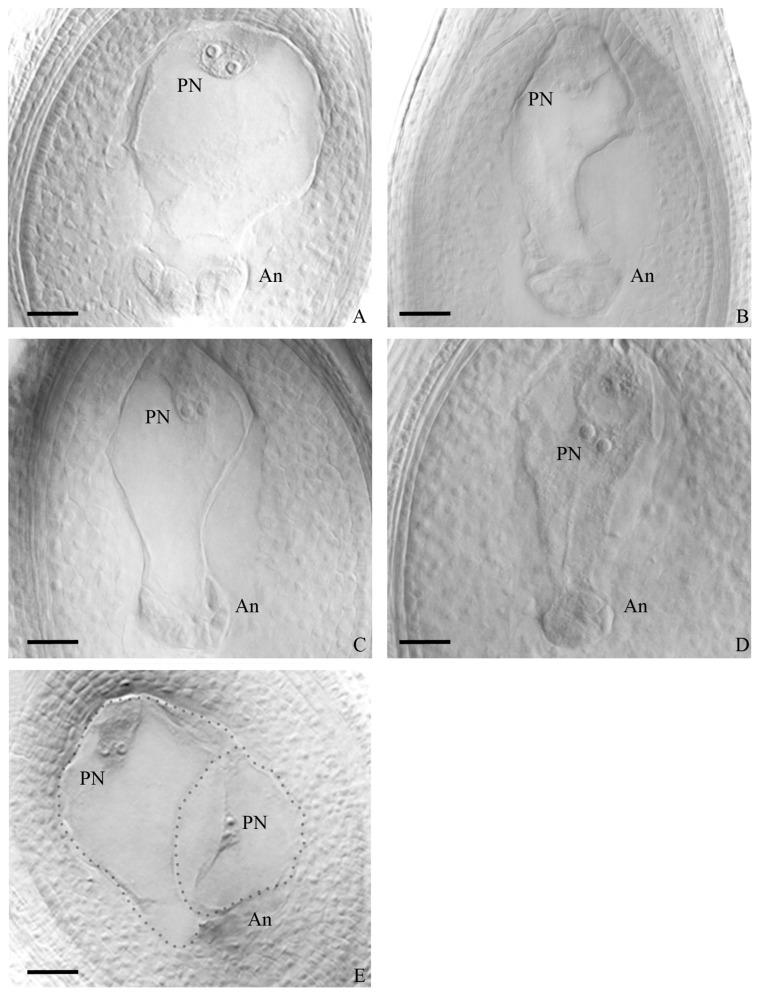
Ovule images showing meiotic embryo sacs having two polar nuclei (PN) at the micropylar end (**top**) and a mass of antipodes (An) at the chalazal end (**bottom**). Egg cells and synergids were visible but in different photographic focus (not shown). *P. durifolium* (**A,E**); *P. ionanthum* (**B**); *P. regnellii* (**C**); *P. urvillei* (**D**). Image (**E**) shows an ovule from *P. durifolium* carrying a typical meiotic embryo sac plus an aposporous embryo sac with two polar nuclei and lacking antipodes. Bar = 100 µm.

**Figure 4 genes-14-01137-f004:**
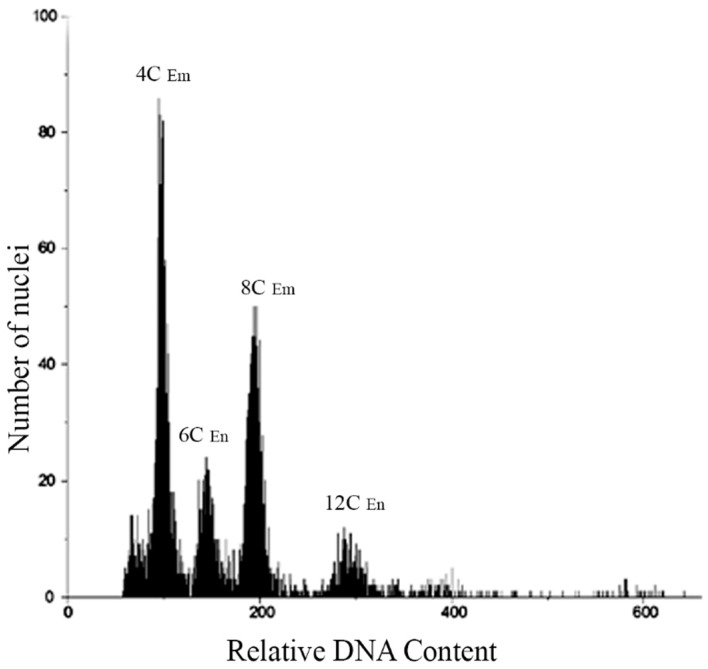
Histogram of cell nuclei from five bulked seeds of a tetraploid plant from the PD3 population of *Paspalum durifolium*. Peaks of relative DNA contents at 100 and 200 correspond to G1 and G2 of tetraploid embryos, while those at 150 and 300 correspond to the endosperm of tetraploid seeds. These peaks show the occurrence of sexual seeds originated by the fertilization of meiotically reduced egg cells and reduced male gametes (Em/En ratio: 2C/3C). Two small peaks of relative DNA content at 400 and 600 correspond to embryo and endosperm, respectively, from a hexaploid seed originated by the fertilization of a non-reduced egg cell and one reduced male gamete (B_III_ hybrid, 2*n* + *n*). Embryo (Em), Endosperm (En).

**Table 1 genes-14-01137-t001:** Identification, collection sites, and the number of plants analyzed of natural populations of *Paspalum* species from northeastern Argentina.

Species	Population Code (Voucher)	Collection Sites	Geographical Coordinates	No. Plants
*P. durifolium*	PD1 (M6)	Corrientes, Dpto. Ituzaingó, NR 12	27°37.622 S, 56°44.873 W	19
	PD2 (M7)	Corrientes, Dpto. San Miguel, NR 118, and PR 5	27°55.901 S, 57°30.996 W	18
	PD3 (M16)	Corrientes, Dpto. Santo Tomé, Esteros del Iberá, PR 40	28°17.435 S, 56°46.219 W	17
	PD4 (M17)	Corrientes, Dpto. Santo Tomé, NR 120	27°49.513 S, 56°15.294 W	18
	PD5 (M18)	Corrientes, Dpto. Ituzaingó, NR 118	27°33.646 S, 57°09.184 W	20
*P. ionanthum*	PI1 (M4)	Corrientes, Dpto. Ituzaingó, NR 120	27°49.265 S, 56°16.328 W	20
	PI2 (M8)	Corrientes, Dpto. Concepción, Santa Rosa, NR 118	28°14.245 S, 58°03.252 W	20
	PI3 (M10)	Corrientes, Dpto. General Paz, Paso Florentín, PR 5	27°45.205 S, 57°45.658 W	16
	PI4 (M11)	Corrientes, Dpto. Mburucuyá, Parque Nacional Mburucuyá.	28°02.024 S, 58°02.172 W	20
	PI5 (M12)	Corrientes, Dpto. Goya, Paraje Marucha, NR 12	29°09.906 S, 59°05.795 W	20
*P. regnellii*	PR1 (M2)	Misiones, Dpto. Montecarlo, Puerto Laharrague, NR 12	26°32.492 S, 54°43.506 W	20
	PR2 (M19)	Misiones, Dpto. San Ignacio, Jardín América, NR 12	27°03.170 S, 55°15.040 W	19
	PR3 (M20)	Misiones, Dpto. San Pedro, NR 14	26°43.626 S, 54°14.845 W	20
	PR4 (M21)	Misiones, Dpto. 25 de Mayo, PR 2	27°22.441 S, 54°25.503 W	18
	PR5 (M22)	Misiones, Dpto. 25 de Mayo, PR 4	27°46.628 S, 55°14.998 W	19
*P. urvillei*	PU1 (M1)	Misiones, Dpto. San Ignacio, 2,2 Km East PR 210	27°16.613 S, 55°27.735 W	17
	PU2 (M5)	Corrientes, Dpto. Ituzaingó, NR 120	27°49.265 S, 56°16.328 W	20
	PU3 (M14)	Entre Ríos, Dpto. La Paz, PR 6	31°01.223 S, 59°25.222 W	20
	PU4 (M23)	Santa Fe, Dpto. Gral. Obligado, NR 11	28°35.733 S, 59°25.005 W	20
	PU5 (M27)	Chaco, Dpto. Gral. San Martin, PR 90	26°24.117 S, 59°22.726 W	17

PD = *P. durifolium*; PI = *P. ionanthum*; PR = *P. regnellii*; PU = *P. urvillei*; M = Martínez E. J. et al.; RN = National route; PR = Provincial route.

**Table 2 genes-14-01137-t002:** Observed number and percentage (%) of embryo sac types, and the observed proportions of the sexual (SP) and apomictic (AP) pathways at the ovule stage in populations of four *Paspalum* species.

Species	Pop	*n*	Number of Ovules Bearing (%)				Proportions		
			MES	AES	MES + AES	AbES	SP	AP	*p*
*P. durifolium*	PD1	152	127 (83.5)	-	1 (0.7)	24 (15.8)	0.99	0.01	<0.001
	PD2	150	141 (94.0)	-	-	9 (6.0)	1.00	0.00	<0.001
	PD3	152	134 (88.2)	-	6 (3.9)	12 (7.9)	0.96	0.04	<0.001
	PD4	153	129 (84.3)	-	10 (6.5)	14 (9.2)	0.93	0.07	<0.001
	PD5	156	124 (79.5)	-	5 (3.2)	27 (17.3)	0.96	0.04	<0.001
*P. ionanthum*	PI1	158	150 (95.0)	-	4 (2.5)	4 (2.5)	0.97	0.03	<0.001
	PI2	151	147 (97.4)	-	2 (1.3)	2 (1.3)	0.99	0.01	<0.001
	PI3	150	149 (99.3)	-	-	1 (0.7)	1.00	0.00	<0.001
	PI4	165	160 (97.0)	-	-	5 (3.0)	1.00	0.00	<0.001
	PI5	158	156 (98.7)	-	-	2 (1.3)	1.00	0.00	<0.001
*P. regnellii*	PR1	160	112 (70.0)	-	-	48 (30.0)	1.00	0.00	<0.001
	PR2	151	132 (87.4)	-	-	19 (12.6)	1.00	0.00	<0.001
	PR3	155	148 (95.5)	-	-	7 (4.5)	1.00	0.00	<0.001
	PR4	170	162 (95.3)	-	-	8 (4.7)	1.00	0.00	<0.001
	PR5	164	153 (93.3)	-	-	11 (6.7)	1.00	0.00	<0.001
*P. urvillei*	PU1	167	160 (95.8)	-	-	7 (4.2)	1.00	0.00	<0.001
	PU2	158	153 (96.8)	-	-	5 (3.2)	1.00	0.00	<0.001
	PU3	156	155 (99.4)	-	-	1 (0.6)	1.00	0.00	<0.001
	PU4	156	151 (96.8)	-	-	5 (3.2)	1.00	0.00	<0.001
	PU5	155	155 (100)	-	-	-	1.00	0.00	<0.001

MES, meiotic embryo sac; AES, aposporous embryo sac; MES + AES, meiotic embryo sac plus aposporous embryo sac; AbES, aborted embryo sac. Significant differences (*p* < 0.05).

**Table 3 genes-14-01137-t003:** Expected (Ei) and observed (Oi) proportions of sexuality and apomixis at the seed stage and the reproductive efficiency of each pathway in populations of four *Paspalum* species.

Species	Pop	Sexual Pathways		Apomictic Pathways		Statistical Analysis		Reproductive Efficiency	
		Ei	Oi	Ei	Oi	χ^2^	*p*	Sex	Apo
*P. durifolium*	PD1	0.99	1.00	0.01	0.00	4.67 × 10^−31^	1.00	1.01	-
	PD2	1.00	1.00	0.00	0.00	-	-	1.06	-
	PD3	0.96	1.00	0.04	0.00	0.333	0.564	1.04	-
	PD4	0.93	1.00	0.07	0.00	1.049	0.3055	1.07	-
	PD5	0.96	1.00	0.04	0.00	0.237	0.626	1.04	-
*P. ionanthum*	PI1	0.97	1.00	0.03	0.00	0.036	0.849	1.03	-
	PI2	0.99	1.00	0.01	0.00	5.5 × 10^−31^	1.00	1.01	-
	PI3	1.00	1.00	0.00	0.00	-	-	1.00	-
	PI4	1.00	1.00	0.00	0.00	-	-	1.00	-
	PI5	1.00	1.00	0.00	0.00	-	-	1.00	-
*P. regnellii*	PR1	1.00	1.00	0.00	0.00	-	-	1.00	-
	PR2	1.00	1.00	0.00	0.00	-	-	1.00	-
	PR3	1.00	1.00	0.00	0.00	-	-	1.00	-
	PR4	1.00	1.00	0.00	0.00	-	-	1.00	-
	PR5	1.00	1.00	0.00	0.00	-	-	1.00	-
*P. urvillei*	PU1	1.00	1.00	0.00	0.00	-	-	1.00	-
	PU2	1.00	1.00	0.00	0.00	-	-	1.00	-
	PU3	1.00	1.00	0.00	0.00	-	-	1.00	-
	PU4	1.00	1.00	0.00	0.00	-	-	1.00	-
	PU5	1.00	1.00	0.00	0.00	-	-	1.00	-

χ^2^, Chi-square statistics, significant differences (*p* < 0.05).

**Table 4 genes-14-01137-t004:** Mean comparative analysis of seed set within and among populations for two pollination methods (self- and open pollination) during two flowering periods (1st and 2nd).

Species	Pop	Seed Set (%)								
		Self-Pollination			Open Pollination			Self- Pollination	Open Pollination	
		1st Period	2nd Period	*p* Value ^α^	1st Period	2nd Period	*p* Value ^α^	Total ^†^	Total ^†^	*p* Value ^β^
*P. durifolium*	PD1	0.27 ^a^	0.61 ^a^	0.281	28.09 ^a^	29.58 ^a^	0.839	0.44 ^a^	28.83 ^b^	**<0.001**
	PD2	0.35 ^a^	0.65 ^a^	0.333	46.30 ^a^	51.54 ^a^	0.326	0.50 ^a^	48.92 ^a^	**<0.001**
	PD3	0.16 ^a^	0.65 ^a^	0.262	57.90 ^a^	45.16 ^a^	0.242	0.41 ^a^	51.53 ^a^	**<0.001**
	PD4	1.24 ^a^	0.47 ^a^	0.409	45.52 ^a^	42.70 ^a^	0.881	0.86 ^a^	44.11 ^ab^	**<0.001**
	PD5	0.17 ^a^	1.18 ^a^	0.119	33.17 ^a^	27.97 ^a^	0.507	0.68 ^a^	30.57 ^b^	**<0.001**
	*p* value ^γ^	0.284	0.701		0.173	0.058		0.74	**0.006**	
*P. ionanthum*	PI1	1.00 ^a^	1.00 ^a^	0.995	35.60 ^a^	23.20 ^a^	0.168	1.04 ^a^	29.41 ^c^	**<0.001**
	PI2	0.26 ^a^	1.22 ^a^	0.418	60.25 ^a^	31.27 ^b^	**0.002**	0.75 ^a^	45.80 ^ab^	**<0.001**
	PI3	4.26 ^a^	1.36 ^a^	0.337	49.66 ^a^	58.88 ^a^	0.661	2.79 ^a^	54.27 ^a^	**<0.001**
	PI4	0.16 ^a^	1.10 ^a^	0.164	52.80 ^a^	39.22 ^a^	0.092	0.61 ^a^	46.02 ^ab^	**<0.001**
	PI5	2.85 ^a^	0.71 ^a^	0.089	45.53 ^a^	27.82 ^a^	**0.036**	1.80 ^a^	36.69 ^bc^	**<0.001**
	*p* value ^γ^	0.565	0.972		0.191	**0.004**		0.574	**0.017**	
*P. regnellii*	PR1	8.51 ^a^	4.96 ^a^	0.440	23.47 ^a^	34.19 ^a^	0.360	6.73 ^b^	28.83 ^b^	**<0.001**
	PR2	25.93 ^a^	6.19 ^b^	**0.018**	38.14 ^a^	44.17 ^a^	0.602	16.06 ^ab^	41.16 ^ab^	**0.001**
	PR3	12.44 ^a^	9.74 ^a^	0.704	24.06 ^a^	28.70 ^a^	0.682	11.09 ^ab^	26.38 ^b^	**0.031**
	PR4	10.48 ^a^	2.34 ^a^	0.160	35.83 ^a^	20.77 ^a^	0.144	6.41 ^b^	28.30 ^b^	**<0.001**
	PR5	25.16 ^a^	12.47 ^a^	0.055	62.20 ^a^	37.35 ^a^	0.103	18.81 ^a^	49.77 ^a^	**0.001**
	*p* value ^γ^	0.059	0.142		**0.006**	0.426		0.039	0.026	
*P. urvillei*	PU1	44.61 ^a^	40.75 ^a^	0.828	80.15 ^a^	59.76 ^a^	0.12	42.68 ^a^	69.96 ^a^	**0.018**
	PU2	30.76 ^a^	52.99 ^a^	0.071	86.87 ^a^	74.41 ^a^	0.269	41.88 ^a^	80.64 ^a^	**<0.001**
	PU3	38.47 ^a^	32.75 ^a^	0.671	92.01 ^a^	64.01 ^b^	**0.007**	35.61 ^a^	78.01 ^a^	**<0.001**
	PU4	33.34 ^a^	52.22 ^a^	0.224	76.71 ^a^	69.56 ^a^	0.428	42.78 ^a^	73.32 ^a^	**0.004**
	PU5	31.80 ^a^	54.44 ^a^	0.057	74.00 ^a^	66.05 ^a^	0.391	43.12 ^a^	70.03 ^a^	**0.002**
	*p* value ^γ^	0.906	0.200		0.091	0.767		0.893	0.483	

Tukey’s test significant values are in bold (*p* < 0.05). ^†^ Overall mean values for either self- or open-pollination condition. ^α^ Within-populations significance level between flowering periods (1st vs. 2nd) for either self- or open-pollination condition. ^β^ Within-populations significance level between pollination conditions (self- vs. open pollination), considering the two flowering periods together. ^γ^ Among-populations significance level for either self- or open-pollination condition. Bold values show significant differences. Different letters in each columns indicate significant differences between populations for each species.

## Data Availability

All the data presented in this study are available in the article and in Appendix A.

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
