# Peer review of "Alternative Evolutionary Pathways in Paspalum Involving Allotetraploidy, Sexuality, and Varied Mating Systems"

_genes, 2023, doi:10.3390/genes14061137_

Round 1

Reviewer 1 Report

In the manuscript entitled: "Alternative evolutionary pathways in Paspalum involving allotetraploidy, sexuality, and varied mating systems" (Number: genes-2409653), the Authors performed research focused on the genetic diversity (cytotype composition) of several species of plants belonging to the genus Paspalum. The assess the reproductive mode under common environmental conditions, and determination of the mating system and fertility for each analyzes species will provide insight into the origin, dynamics and adaptation of cytotypes within populations growing in the wild. The presented research shows a large genetic diversity within the Paspalum genus and the study represent important contribution to understand biodiversity in broad context.

In conclusion, I think that the scientific value of the studies presented in the manuscript is high. In addition, the research data included in the manuscript are clearly described and meet the standards. There are only few errors or inaccuracies in the text, which I have listed below.

The scientific standard in embryological research requires that the embryo sac should be presented in the micropylar-chalazal axis, with the micropylar pole on top (even when the ovule is not orthotropic). Therefore, I am asking you to verify the arrangement of the photographs in Figure 2. I think that in the presented data, the images from the karyotype analyzes of cells from the root of the species are missing, and should be added to meet the accepted standards in data presentation.

Author Response

Letter of response to comments and suggestions stated by Reviewer 1

Reviewer comment or suggestions:

  1. The scientific standard in embryological research requires that the embryo sac should be presented in the micropylar-chalazal axis, with the micropylar pole on top (even when the ovule is not orthotropic). Therefore, I am asking you to verify the arrangement of the photographs in Figure 2.

Response:

Thank you for your valuable feedback on our manuscript. We appreciate your attention to detail and your suggestion regarding Figure 2. We have carefully considered your comment and made the necessary modifications accordingly.

As per your suggestion, we have adjusted the arrangement of the photographs in Figure 2 to adhere to the scientific standard in embryological research. Specifically, we have ensured that the embryo sac is now presented in the micropylar-chalazal axis, with the micropylar pole positioned on top.

To verify the changes made, we kindly invite you to review the revised Figure 2, which can be found on Page 10 of the revised version of the manuscript. We believe that the new arrangement better aligns with the required scientific presentation standards in embryological research.

  1. I think that in the presented data, the images from the karyotype analyses of cells from the root of the species are missing and should be added to meet the accepted standards in data presentation.

Response:

We appreciate your suggestion regarding the inclusion of karyotype analysis images of root cells from the studied Paspalum species. We have taken your comment into consideration and have addressed this concern in the revised version of the manuscript. 

As per your recommendation, we have added the images of karyotype analyses to the revised manuscript. These images, depicting the karyotypes of cells from the roots of the four Paspalum species under investigation, are now presented as Figure 2, following Figure 1, on Page 8.

By incorporating these additional images, we believe that we have fulfilled the accepted standards in data presentation, providing a more comprehensive and informative representation of our research findings.

Note: We would like to inform you that in addition to addressing the comments and suggestions, we have also made several stylistic corrections in the manuscript. These changes have been detailed with track changes in the revised version of the document in MS Word.

Reviewer 2 Report

- Figure 1, it seems that the events to the left (usually debris) of the peaks were trimmed. It would be better if the figures showed all the events (as in Figure 2); so that readers can see that, the population of nuclei is clearly distinguished from the other particles to the left and right of the peak. The authors may have a look on Figure 1 in Karafiatova et al. (2020), Doi: 10.1093/jxb/eraa548, Figure 1 in Said et al. (2019), Doi: 10.3835/plantgenome2018.12.0096 and Figure 5 in Said et al. (2022), Doi: 10.3389/fpls.2022.1017958.

- In this study, of the four species all the analyzed populations were detected tetraploid. However, other researchers reported the existence of other ploidies, such as diploids, hexaploids, octaploids…etc. I think ti would be better if the authors provide some data on the percentage (%) of the different ploidies (i.e. diploid, tetraploid, hexaploid...etc) in nature. This would justify the degree of expectation of finding them or not in a specific area. Perhaps the tetraploid is highly dominant, while the other ploidies are very rare in nature and hard to find.

Author Response

Letter of response to comments and suggestions stated by Reviewer 2

Reviewer comment or suggestions:

  1. Figure 1, it seems that the events to the left (usually debris) of the peaks were trimmed. It would be better if the figures showed all the events (as in Figure 2); so that readers can see that, the population of nuclei is clearly distinguished from the other particles to the left and right of the peak. The authors may have a look on Figure 1 in Karafiatova et al. (2020), Doi: 10.1093/jxb/eraa548, Figure 1 in Said et al. (2019), Doi: 10.3835/plantgenome2018.12.0096 and Figure 5 in Said et al. (2022), Doi: 10.3389/fpls.2022.1017958.

Response:

Thank you for your valuable feedback on our manuscript. We appreciate your suggestion regarding Figure 1 and have carefully considered your comment. Based on your recommendation, we have made the necessary modifications to the figure to better align with the requested presentation style.

In the revised version, we have ensured that all the events to the left of the peaks, including debris, are now included in Figure 1. By including these events, we aim to provide readers with a clearer understanding of how the population of nuclei is distinguished from other particles surrounding the peaks.

  1. In this study, of the four species all the analysed populations were detected tetraploid. However, other researchers reported the existence of other ploidies, such as diploids, hexaploids, octoploids…etc. I think it would be better if the authors provide some data on the percentage (%) of the different ploidies (i.e. diploid, tetraploid, hexaploid...etc) in nature. This would justify the degree of expectation of finding them or not in a specific area. Perhaps the tetraploid is highly dominant, while the other ploidies are very rare in nature and hard to find.

Response:

Thank you for your insightful comment and suggestion regarding the ploidy levels in the investigated Paspalum species. We appreciate your interest in understanding the distribution of different ploidies in nature and the potential dominance of tetraploids.

We acknowledge that most studies on chromosome counts in the Paspalum species under investigation have been conducted on a limited number of plants, making it challenging to provide precise percentages of different ploidy levels for each species. Due to the limited sample size in previous studies, the percentages of ploidy levels found for a specific species may not be representative or statistically significant.

However, in our revised manuscript, we have included additional information in the Discussion section, on page 15, where we present the observed frequency of each ploidy level when multiple plants of the same species were analysed. While these data may not provide comprehensive percentages for the overall distribution of ploidy levels in nature, they offer insights into the relative frequency of different ploidies within the analysed plant populations.

We believe that by presenting the observed frequencies of different ploidy levels for each species, we contribute to a better understanding of the prevalence and rarity of various ploidies in nature.

Note: We would like to inform you that in addition to addressing the comments and suggestions, we have also made several stylistic corrections in the manuscript. These changes have been detailed with track changes in the revised version of the document in MS Word.
